



**Differentiating between particle formation and growth events in an urban**
2                                      **environment**

Buddhi Pushpawela, Rohan Jayaratne, and Lidia Morawska[*]
International Laboratory for Air Quality and Health, Queensland University of Technology,
GPO Box 2434, Brisbane, QLD 4001, Australia
**\*** Corresponding author contact details: Email: l.morawska@qut.edu.au
**Abstract**
Small aerosols at a given location in the atmosphere often originate in-situ from new particle formation
(NPF). However, they can also be produced and then transported from a distant location to the point of
observation where they may continue to grow to larger sizes. This study was carried out in the
subtropical urban environment of Brisbane, Australia, in order to assess the relative occurrence
frequencies of NPF events and particle growth events with no NPF. We used a neutral cluster and air
ion spectrometer (NAIS) to monitor particles and ions in the size range 2-42 nm on 485 days, and
identified 236 NPF events on 213 days. The majority of these events (37%) occurred during the daylight
hours with just 10% at night. However, the NAIS also showed particle growth with no NPF on many
nights (28%). Using a scanning mobility particle sizer (SMPS), we showed that particle growth
continued at larger sizes and occurred on 70% of nights, typically under high relative humidities. Most
particles in the air, especially near coastal locations, contain hygroscopic salts such as sodium chloride
that may exhibit deliquescence when the relative humidity exceeds about 75%. The growth rates of
particles at night often exceeded the rates observed during NPF events. Although most of these night
time growth events were preceded by daytime NPF events, the latter was not a prerequisite for growth.
We conclude that particle growth in the atmosphere can be easily misidentified as NPF, especially when
they are monitored by an instrument that cannot detect them at the very small sizes.
**Keywords**: New particle formation, particle growth, atmospheric aerosols, secondary particles.
**1 Introduction**
The formation of secondary particles in the atmosphere through homogeneous nucleation is known as
new particle formation (NPF). This is one of the major sources of particles in the atmosphere. The
condensable species that contribute are mainly sulfuric acid and semivolatile organic compounds and
the process is thought to occur by binary water-sulfuric-acid or ternary water-sulfuric-acid-ammonia
nucleation. Particles, thus formed, form stable clusters that continue to grow to larger sized particles by
vapour condensation or by coagulation with other particles (Kulmala et al., 2013).
The particle formation rate and the particle growth rate are the two most important parameters used to
characterize an NPF event. The particle formation rate is the rate of formation of smallest measurable
size of the particles, generally in the size range 2-3 nm. This is different to the actual nucleation rate
(the rate at which the stable clusters form). The particle growth rate varies with particle size and, hence,
the reported values depend on the detectable size ranges of the instruments used. Until recently, studies
have been limited to measure the particles above 3 nm. However, it is only during the past decade that
the advancement of instruments has developed to such a level that particles of 2 nm or even smaller can
be measured (Kulmala et al., 2012).

NPF has been observed under a range of environmental conditions, on every continent in the world
(Kulmala et al., 2004). The occurrence rate of NPF is mainly dependent on the nature and concentration
of gaseous precursors, which are controlled by a number of factors including the type and intensity of
the sources, concentration of pre-existing aerosols, origin of air masses, photo-chemical processes and
meteorological parameters such as intensity of solar radiation, temperature, relative humidity, wind
direction and wind speed (Birmili and Wiedensohler, 2000). Pre-existing aerosols act as sinks to
condensable gases that are present in the atmosphere. This leads to a reduction in their vapour pressure
and inhibits homogeneous nucleation.

Oxides of nitrogen and volatile organic compounds are readily produced in urban environments from
sources such as motor vehicles and industrial facilities. These gases react with ozone in the presence of
sunlight to produce OH radicals that can oxidise gaseous precursors such as sulphur dioxide and nitric
oxide, converting them into the condensable species sulfuric acid and nitric acid, respectively. These



photochemical reactions are more likely to occur during the day time on sunny days with high intensity
of solar radiation, which is when we would expect to observe more NPF events.

Numerous studies in many different environments have conclusively shown that the large majority of
NPF occur during the day time. Very few studies have reported the occurrence of NPF during the night
time and these have mostly been in forest environments and coastal sites. Table 1 gives a summary of
studies in chronological order, that have reported observations and frequencies of occurrence of night
time NPF events, together with the respective frequencies of occurrence of day time NPF events and the
instrumentation that was used. We see that, at a given location, NPF events were generally more likely
to occur during the day time than during the night. The sole exception is the short study of 16 days by
Kammer et al. (2017). Night time events were reported on between 4% and 37% of the days observed.
They were more likely to be observed at forest locations (16% to 37%), while the two studies conducted
at coastal locations showed significantly lower values of 4% and 11%. In a previous study carried out in
and around Brisbane with an SMPS, Salimi et al. (2017) reported NPF events on around one in every
four nights. They also reported NPF on every second day which is significantly higher than any of the
values found in Brisbane (Guo et al., 2008;Cheung et al., 2011;Crilley et al., 2014;Jayaratne et al.,
2016;Pushpawela et al., 2018).

In the present study, we collected data of charged and uncharged particle concentrations in the urban
environment of Brisbane using a neutral cluster and air ion spectrometer (NAIS) on close to five
hundred days. The NAIS can provide more accurate information on NPF than the SMPS, because of its



ability to measure particles down to 2 nm in size, which is very close to the size at which the initial
steps of nucleation and formation of particles occur (Manninen et al., 2011;Manninen et al., 2016). The
results were compared with that obtained simultaneously with an SMPS with a minimum detectable size
of 9 nm. The SMPS data were also used to determine the growth rates of particles. The observations by
the NAIS and SMPS were used to differentiate between (a) local NPF events followed by particle
growth and (b) growth events in the absence of NPF events – two phenomena that are not always
concurrent and often misidentified when only one instrument is used.



## 2 Methods

### 2.1 Monitoring Site

The instruments were housed in a sixth-floor laboratory in a building at the Gardens Point campus of the Queensland University of Technology in Brisbane, Australia. The site is situated at the edge of the Brisbane Central Business District bordered by the City Botanical Gardens and the Brisbane River, approximately 100 m away from a busy motorway carrying about 120,000 vehicles per day and is representative of a typical urban environment in Australia. The measurements were carried out during the three calendar years 2012, 2015 and 2017, yielding 485 complete days of data.

The pollutants at this site were mainly from motor vehicle exhaust emissions. Depending on the wind direction, emissions may also be received from the Port of Brisbane and two oil refineries in its vicinity as well as from Brisbane Airport, all located about 20 km to the north-east of the monitoring site.

Meteorological data such as temperature, relative humidity, solar radiation, rainfall, wind direction and wind speed as well as air quality data such as sulphur dioxide ($SO_2$), ozone ($O_3$), $PM_{10}$, $PM_{2.5}$ and atmospheric visibility were obtained from the Department of Environmental and Heritage Protection, Queensland, at their in-situ site at the Queensland University of Technology and two other sites within a distance of 1.5 km from the University.



## 2.2 Description of the instruments

The NAIS, manufactured by Airel Ltd, Estonia (Manninen et al., 2016), detects the mobility distribution of charged clusters and particles of both polarities in the electrical mobility range from 3.2 to 0.0013 $cm^2$ $V^{-1}s^{-1}$. It also measures the size distribution of total particles in the size range from 2.0 - 42 nm. The instrument has a high-resolution time down to 1 s and consists of two cylindrical electrical mobility analysers, one for each polarity. It operates in four modes: ion mode; particle mode; alternate charging mode and offset mode. In the ion mode, the NAIS measures naturally charged particles without any modification. In the particle mode, it measures all charged and uncharged particles. The lower detection limit in the particle mode is restricted to 2 nm due to presence of corona generated ions in the instrument (Manninen et al., 2016). The alternate charging mode is similar to the particle mode, but it electrically neutralizes the sampled particles and improves the performance of the instrument. In the offset mode, the NAIS measures zero signals, noise levels and parasitic currents. The measurement process of the instrument is fully automated. The measurement cycle of the NAIS varies from 2-5 minutes. A more detailed discussion of its design and principles is given in (Manninen et al., 2011) and (Mirme and Mirme, 2013). In this study, we set the measurement cycle to 2 min ion mode, 2 min particle mode, and 1 min offset mode.

An SMPS, consisting of a TSI model 3071 differential mobility analyser and a TSI model 3782 condensation particle counter, was used to measure the particle size distribution in the range from 9 - 415 nm.



## 2.3 Data Analysis

### 2.3.1 Classification of New Particle Formation (NPF) events:

We identified NPF events using the rate of change of total particle concentration, dN/dt, where N is the number of particles in the size range 2.0 -10.0 nm and using the classification described by [Zhang et al 2004]. Events with N > 10,000 $cm^{-3}$ for at least 1 hour and dN/dt >10,000 $cm^{-3}h^{-1}$ were defined as "strong" NPF events. Events with 5000 < N < 10,000 $cm^{-3}$ for at least 1 hour and 5000 < dN/dt < 10,000 $cm^{-3}h^{-1}$ were classified as "weak" NPF events. All of these events started in the nucleation mode size range and prevailed over a time span of more than one hour, generally exhibiting a "banana" shape in the time-series contour plot of particle number concentration (PNC), indicating particle formation and subsequent growth. A 24-hour day that included at least one NPF event was labelled as an 'NPF Day'. A day on which there were no NPF events was labelled as a 'Non-event Day'.

Every NPF event was characterised by a sharp increase of the PNC in the intermediate size range from 2.0-7.0 nm. This observation has been used to determine the starting time of an NPF event (Leino et al., 2016). Similarly, in the present study, the starting time of a strong NPF event was determined by noting the time of first occurrence of dN/dt > 10,000 $cm^{-3}h^{-1}$. The starting time of a weak NPF event was determined by noting the time of first occurrence of dN/dt > 5000 $cm^{-3}h^{-1}$. N is the number of particles in the size range 2.0-10 nm.

NPF events that started between 6:00 am and 6:00 pm were categorized as "day time" NPF. NPF events that started between 6:00 pm and 6:00 am were categorized as "night time" NPF.



### 2.3.2 Classification of Growth events:


The data from the NAIS showed that growth events were not always preceded by an NPF event. Growth
events that did not follow an NPF appeared as a "floating-banana" shape in the PNC contour plots.
These events were identified using the rate of change in the diameter ($d_p$) of particle, $dd_p/dt$. Events with
$dd_p/dt > 1$ nm h$^{-1}$ were classified as "growth" events. In the NAIS data, these events showed an
enhancement of PNC in the size range above 7 nm. Further, in these events, unlike in NPF events, the
sharp increase in PNC in the size range between 2-7 nm was absent. In this way, growth events could be
clearly distinguished from NPF events. In fact, unless they were preceded by an NPF event, most
growth events showed very few particles in the size range below 10 nm. We also observed "vertical
band" shapes which were due to the sudden appearance of high concentrations of particles in all sizes.
These were neither NPF nor growth events and characterised the influx of already formed particles from
further locations to the monitoring site, and were ignored in the analysis.

### 2.3.3 Calculation of particles growth rate


The growth rate (GR) of particles is defined as
$$GR = \frac{dd_p}{dt} = \frac{d_{p2}-d_{p1}}{t_2-t_1} \tag{1}$$

where $dp_2$ and $dp_1$ are the diameters of particles at times $t_1$ and $t_2$. This was calculated by the maximum
concentration method described in (Kulmala et al., 2012). The unit of the GR is nanometres per hour.
During an NPF or a growth event, the number concentration of small particles increases, showing a



peak in the particle size distribution. When the particles grow in size, this peak shifts towards larger
sizes. In order to derive the maximum particle concentration, we plotted the time series of the PNC in
different size ranges. We estimated the GR from the slope of the best-fitted line on the graph of mid-
point diameter of particles versus the time of maximum concentration (Dos Santos et al., 2015;Pierce et
al., 2014).
**2.3.4 Statistically significant differences**
Statistical significances of the difference between two parameters were calculated using the Student's t
test.



## 3. Results and Discussion

### 3.1 Observation of NPF during study period

The study yielded complete 24h data on a total of 485 days. The instrument was unavailable on some days, as it was required for other projects or was being serviced or cleaned. In addition, a few days were 'lost' due to missing data owing to power failures or instrument malfunction. A summary of the observational periods, together with the corresponding number of days on which 24h data were available and NPF events were observed, is shown in Table 2. Columns 3 to 8 represent the number of day time, night time and total NPF classified into strong and weak events according to the method described in section 2.3.1. The last three columns give a summary of all NPF events.

Altogether, 236 NPF events (strong and weak) were observed on 213 of the 485 days on which we were able to obtain data. Out of this, strong NPF events were observed on 177 days, giving an occurrence rate of 37%. This is only slightly less than the rate of 41% found by Pushpawela et al. (2018) using the NAIS in Brisbane over the single calendar year 2012. In the two other studies using the NAIS in Brisbane, Crilley et al. (2014) and Jayaratne et al. (2016) reported higher values of 56% and 45% respectively. However, both these previous studies used a slightly different criteria to identify NPF events, that is they excluded the requirement of $N > 10,000$ cm$^{-3}$ for a period of at least 1 hour. The Crilley et al. (2014) study was also conducted over a much shorter period of 36 days only. Table 2 also shows that, although "strong" day time NPF events were observed on 159 days (33%), "strong" night





time NPFs were relatively scarce, occurring on just 18 days (4%). Further, "weak" NPF events were
observed on 59 days (12%) and these were almost equally distributed between night and day times.
Taking into account all strong and weak NPF, day time NPF occurred on 37% of the days while night
time NPF occurred on only 10%. In Table 2, it should be noted that a given day may sometimes have
both a day time and a night time event. There were 23 such days. In addition, there were 8 days that had
two daytime events and no instances of two events during the same night. There have been three
previous studies that have used an SMPS to study NPF in Brisbane. Together with the occurrence rates
in parenthesis, these were Guo et al. (2008) (35%), Cheung et al. (2011) (26%) and Salimi et al. (2017)

207   (77%).


**3.2 Diurnal variation**

Figure 1 (a) shows a summary of starting times of all NPF events, estimated by using the method
described in Section 2.3.1. Figure 1 (b) shows the histogram of number of events observed in each 30
min period of the day. Both of these figures show that most NPF events (73%) began during the
morning, with a high likelihood of occurrence between 8.00 am and 10.00 am. In particular, 85 out of
236 events occurred during this 2-hour period. This is likely to be a result of several factors such as the
higher concentration of precursor gases from motor vehicles during the morning rush hour and the onset
of solar radiation. However, no NPF were observed during the evening rush hour period around 4-6 pm.
During this time, the air temperatures are still relatively high and, although the gaseous precursors are
being produced, the vapour pressures may not be sufficiently high to produce secondary particles. The



starting times of night time NPF events also showed a distinct trend with a peak likelihood between 8
and 9 pm. By this time of the day, the temperatures have generally fallen sufficiently for vapour
pressures to increase. No night time events were observed at all during the second half of the night,
between 11 pm and 4 am. Although the temperatures are low during this time, there is minimum
production of precursor gases.

**3.3 Effect of atmospheric parameters**

A summary of the mean and range of various meteorological and air quality parameters during NPF and
non-event days is shown in Table 3. The mean solar radiation intensity on NPF days were significantly
higher compared to the other days with mean values of 505 W m$^{-2}$ and 397 W m$^{-2}$, respectively.
Conversely, the mean relative humidity on NPF days was significantly less than on other days with
values of 54% and 66%, respectively. The mean relative humidity on NPF days were 59% and 52%
during winter and summer months. Therefore, NPF events were more likely to occur on days with low
relative humidity and high solar radiation. Similar observations have been reported from several other
urban cites such as Melpitz, Germany (Birmili and Wiedensohler, 2000), San Pietro Capofiume, Italy
(Hamed et al., 2007) and Pune, India (Kanawade et al., 2014).

The wind direction on NPF days was mainly from the south to southwest directions, with a mean wind
speed of around 0.5 m s$^{-1}$. The mean air temperature was 17$^{0}$C and 24$^{0}$C on NPF days during winter and
summer months. We did not detect any clear differences in wind direction, wind speed and air





temperature between NPF days and the other days. In general, most of the NPF events occurred on days

when there was no rainfall observed. However, a clear dependence was found between NPF occurrence

and atmospheric visibility. The visibility was expressed through the particle back scatter coefficient

(BSP) in units of $Mm^{-1}$. These two parameters are inversely proportional to each other. The BSP

observed at 8 am on NPF days was significantly lower on NPF days than on other days, with mean

values of 18 $Mm^{-1}$ and 31 $Mm^{-1}$, respectively. A good discussion about the relationship between the

occurrence of NPF in Brisbane and the values of BSP may be found in Jayaratne et al. (2015). This

study also found that, no NPF events occurred on days when the mean $PM_{2.5}$ exceeded 20 µg m$^{-3}$ in

Brisbane.

The presence of high concentration of $O_3$ under high solar radiation increases the production of OH

radicals, and the presence of high concentration of both $SO_2$ and OH radicals give rise to increased

production of $H_2SO_4$ leading to NPF (Seinfeld and Pandis, 2006;Lee et al., 2008). Therefore, we would

expect $SO_2$ and $O_3$ concentration levels to be higher on NPF days than on non-event days. However, we

observed only a marginal increase of $SO_2$ and $O_3$ concentrations on NPF days (Table 3).

**3.4 Day time and night time NPF events**

The two upper panels in Figure 2 show the NAIS spectrograms obtained between 8:00 am and 4:00 pm

on 19 August 2017 and 31 July 2015, respectively. On 19 August, a strong NPF event began in the

morning at around 9:00 am and lasted for 4-5 hours. Here, the total PNC increased from about 30,000



cm$^{-3}$ at 9:00 am to just over 90,000 cm$^{-3}$ at 11:00 am, giving a particle formation rate of 30,000 cm$^{-3}$ h$^{-1}$.
Thereafter, particles continued to grow in size for several hours. The PNC decreased gradually in the
afternoon. The particles showed a relatively high growth rate of about 7 nm h$^{-1}$ in the size range 2-42
nm.

The two lower panels in Figure 2 show NAIS spectrograms obtained during the night, between 6:00 pm
and 2:00 am on 20 August 2015 and 5 September 2015, respectively. On 20 August, a strong NPF event
began in the night at around 9:30 pm and lasted for 2-3 hours. The particles also showed a relatively
high growth rate of about 11 nm h$^{-1}$ in the size range 2-42 nm.

We did not observe a significant difference in growth rates of particles between daytime and night time
NPF events. Typically, the growth rates were high during the first few hours and then decreased to a
few nanometres per hour within 3-4 hours after nucleation. The growth rate of particles in the size range
2-42 nm during all NPF events, calculated from equation (1), varied between 4 nm h$^{-1}$ and 22 nm h$^{-1}$
with a mean and standard deviation of (12.1 ± 6.5) nm h$^{-1}$.

These growth rates were comparable to the values reported at two other urban locations; Atlanta, USA
(3-20 nm h$^{-1}$)(Stolzenburg et al., 2005) and Budapest, Hungary (2-13 nm h$^{-1}$) (Salma et al., 2011).
However, the mean values of growth rates obtained by previous studies in Brisbane were significantly
lower than the value reported by this study. For example, Cheung et al. (2011) and Salimi et al. (2017)
reported growth rates of 4.6 nm h$^{-1}$ and 2.4 nm h$^{-1}$ respectively. Both these studies were carried out
using an SMPS with a lower detection size of about 10 nm.



## 3.5 Observations of growth events during the study period

NPF events are almost always followed by particle growth. However, with the NAIS, we observed several growth events that were not preceded by an NPF event. These events were observed more often at night than during the day. A summary of these events observed by the NAIS, is shown in Table 4. Columns 3 to 5 represent the number of day time, night time and total growth events classified according to the method described in section 2.3.2. Figure 3 shows examples of NAIS spectragrams of such growth events that occurred during the day time (a) and night time (b). Particle growth is again demonstrated by the typical banana shape of the colour contours, with the difference that the lower end of the 'banana' does not reach as far as the smallest particle sizes, indicating that there is no NPF. This shape is sometimes referred to as a "floating banana", to differentiate it from the complete "banana" shape of an NPF event. In most of the events, particle growth is observed to continue for several hours. The observed rates of growth varied between 1 nm h$^{-1}$ and 45 nm h$^{-1}$ with a mean and standard deviation of (16.8 ± 11.9) nm h$^{-1}$ in the size range 8-42 nm. During the 485 days of observation, excluding NPF events, day time growth events were observed on just 54 days (11%), whereas night time growth events were observed on 135 days (28%). The overall occurrence rate of growth events obtained by the NAIS was 37%. However, it should be noted that particles continued to grow at sizes larger than the upper size detection rate of the NAIS, which was 42 nm. Thus, the SMPS was likely to detect many more growth events than the NAIS.



## 3.6 Observations of particle growth by SMPS

Next, we look at the behaviour of total PNC and the median particle diameter of NPF and growth events

using the data obtained by the SMPS. Figure 4 shows a period of 6 days, during which there were 3

consecutive daytime NPF events that were followed by two non-event days and a day with a daytime

NPF event. The NPF events are shown by red arrows. In each of these four cases, prior to the inception

of the daytime NPF, the total PNC was low - about 2500 $cm^{-3}$. During the NPF event, the total PNC

increased from about 5000 $cm^{-3}$ in the morning to over 15,000 $cm^{-3}$ near mid-day. Thereafter, the

particles started to grow in size up to 20-30 nm. During and after the late afternoon, although the total

PNC began to decrease, the particles continued to grow in size up to 40-65 nm. All 4 NPF events

continued through this "second phase of particle growth" until the early morning of the next day. The

growth rate varied between 2-7 $nm\,h^{-1}$.

Figure 5 shows another example. During this 7 day period, two growth events in the late afternoon were

preceded by NPF events. The remaining two growth events did not follow any NPF event. The particles

grew up to 40-50 nm. During the measurement period, particle growth events were observed on 65-

70% of the nights.

Continued growth of particles following NPF events have been reported by other researchers. For

example, Man et al. (2015) observed 12 out of 17 NPF events with particle growth from 10 nm to 40

nm during the day time at a suburban coastal site in Hong Kong. In addition, they observed 3 events

with second phase of particle growth to 61-97 nm at night time. These three events were preceded by a





daytime NPF event. Russell et al. (2007) observed nanoparticle growth on 19 out of 48 days (40%)
during the day time and on 5 out of 48 days (10%) during the night time in Appledore Island, Maine,
USA. Subsequently, particle growth continued over several hours with rates varying from 3 to 13 nm h⁻
¹.

NPF generally occur at high solar radiation, high temperature and low relative humidity. However,
growth events were more likely to occur during time periods with low temperature and high relative
humidity. We investigated this further by plotting the median particle size and relative humidity as a
function of time during growth events (Figure 6). In general, progression into the night time, after 6:00
pm, was accompanied by a decrease in air temperature, resulting in an increase in relative humidity in
the atmosphere.

During the event that occurred on July 16, 2012 the median particle size increased from about 30 to 65
nm as the relative humidity increased from 65% to 80% (Figure 6a). Similarly, during the event that
occurred on July 20, 2012 the median particle size increased from about 30 to 75 nm as the relative
humidity increased from 75% to 90% (Figure 6b).

It is well-known that atmospheric aerosol particles change their size with relative humidity due to the
uptake of water (Winkler, 1988). Water uptake is caused by the deliquescence of soluble salts which
form a solution when the solid compound is exposed to water vapour at sufficiently high vapour
pressure. Several organic materials are also known to absorb water at high humidity which is more



generally known as hygroscopicity. Figure 7 shows the diameter of a sodium chloride (NaCl)-bearing
particle as a function of relative humidity (Wise et al., 2007). The red line corresponds to the
deliquescence point for NaCl at 76% relative humidity. At this point, the particle deliquesces and
becomes a solution of droplet with a well-defined spherical shape. The particle diameter does not
change considerably as the relative humidity is increased from 0 to 74%. As the relative humidity
increased from 76% to 91%, the particle diameter increased by a factor of 1 or more. Therefore, as the
relative humidity increases, the particles sizes increase due to their affinity to absorb water. Close to the
coast, sea-salt aerosols constitute a large proportion of the atmospheric particulate mass and NaCl is a
major component. Many of the inorganic substances that readily absorb water, such as sea salt,
ammonium salts and nitrates, are present in the Brisbane environment (Harrison, 2007). Therefore, it is
not surprising that, in the present study, we observed that particle growth occurred on 7 out of 10 nights
with high relative humidity.

**3.7 Probability of growth events being misidentified as NPF events**

In Figure 3 (a), the horizontal white line indicates the typical lower size detection threshold of the
SMPS that has been used in many locations before; we chose 7 nm as a typical value in this case. The
SMPS does not 'see' any particles below this line. It is clear that there is an enhancement of PNC in the
size range 7-20 nm around 11:30 am on this day. The absence of intermediate size particles (between 2-
7 nm) suggests that the 7-20 nm particles originated on-site by primary emission or were advected to the
site from a distant location. The NAIS clearly shows that this was not an NPF event. However, in the



absence of information below a particle size of 7 nm, the SMPS data may be easily misinterpreted as an
NPF event. The typical 'floating banana' shape of the spectrogram contours show that the particles
continue to grow between 11:30 a.m. and about 1:00 p.m. and this can be observed by an SMPS. As we
have demonstrated, growth events are not always formation events. There are two enhancement events
near 1.00 pm and 3.30 pm. Once again, the NAIS shows that neither of these are NPF events, although
based on the SMPS they may be mistakenly identified as such. Figure 3 (b) shows another event that
can be easily misidentified as an NPF event based on SMPS data alone.

Salimi et al. (2017), using an SMPS with a lower size limit of 9 nm at 25 sites across Brisbane, reported
219 NPF events out of 285 days of measurements. This occurrence rate of 77% (67% of day time and
33% of night time) is significantly higher than any of the values found previously in Brisbane and at
any other location in the world. With the NAIS, it was possible to show that most of these events were
growth events and not NPF events. It was not possible to differentiate these two types of events with the
SMPS alone as it provides no knowledge of the PNC below 9 nm. With the NAIS, we did not observe
nocturnal NPF events on more than 47 of 500 days.

In many NPF events, particle growth ceases after they have grown to a certain size. In the growth event
in Figure 3, the maximum size is about 25 nm. In such cases, the greater part of the 'banana' profile is
below 7 nm and, thus, invisible to the SMPS. This could result in the missing of such NPF events.





Considering, all the factors above, it is clear that the NAIS has a distinct advantage over the SMPS in correctly identifying NPF events in the atmosphere.

## 4. Summary and Conclusions

We monitored charged and neutral PNCs in the size range 2-42 nm on nearly 500 days over three calendar years in the urban environment of Brisbane, Australia, using a NAIS. The data were used to differentiate between NPF events and growth events with no NPF. Day time NPF events were observed on 37% of the observational days, with night time events on only 10% of the days. NPF events were more likely to occur on days with low relative humidity and high solar radiation. 73% of NPF events occurred during the morning, with the highest probability of occurrence between 8.00 am and 10.00 am. Most of the night time events occurred between 8.00 pm and 9:00 pm. No night time events were observed between 11.00 pm and 4.00 am. 28% of the particle growth events that occurred at night were not preceded by an NPF event. These events were characterized by high growth rates of up to 45 nm h$^{-1}$. The SMPS results showed that particle growth continued at larger sizes from ~40 nm to 70 nm and occurred on 70% of nights. Maximum relative humidities were over 80% on most of these nights. These results show that, when particles are monitored by an instrument such as the SMPS that cannot detect them at the very small sizes, particle growth in the atmosphere may be easily misidentified as NPF, leading to an overestimation of the frequency of the latter.



## Acknowledgements


We are thankful to the Department of Environmental and Heritage Protection, Queensland, for
providing some of the meteorological data used in this study.

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



**Figures**

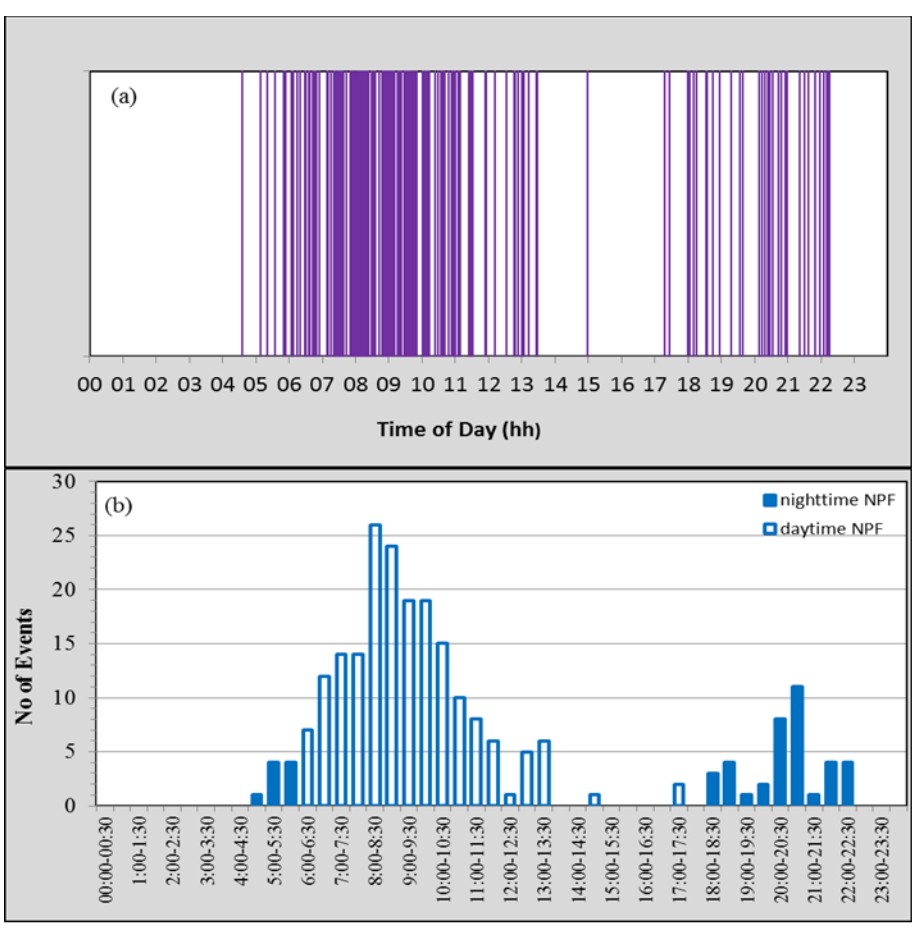


Figure 1: (a) Summary of starting times of all NPF events and (b) histogram for the number of events





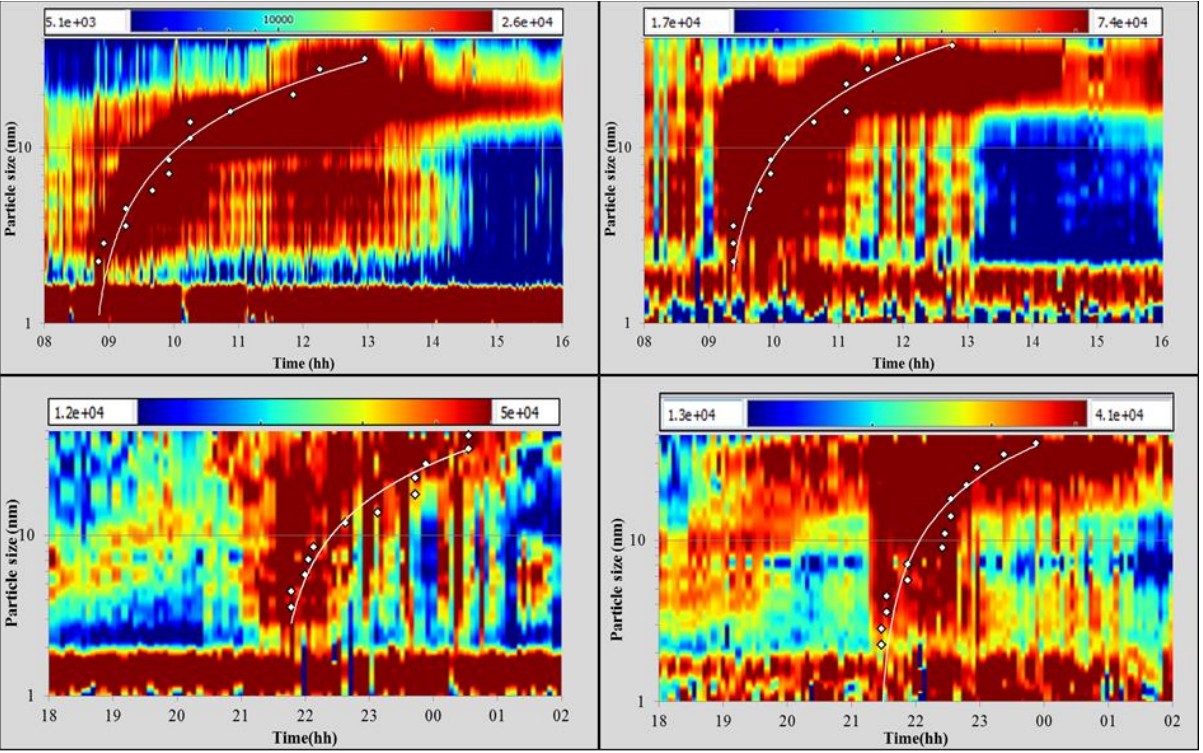

Figure 2: NAIS spectragrams of the daytime NPF events (upper panel) and nighttime NPF (lower panel). The colour contour represents the PNC and the markers represent the times at which the PNC reached its maximum value at each particle size. The unit of PNC is per cubic centimetre. Data below 2 nm should be treated with caution due to instrumentation limitations as described in the text.






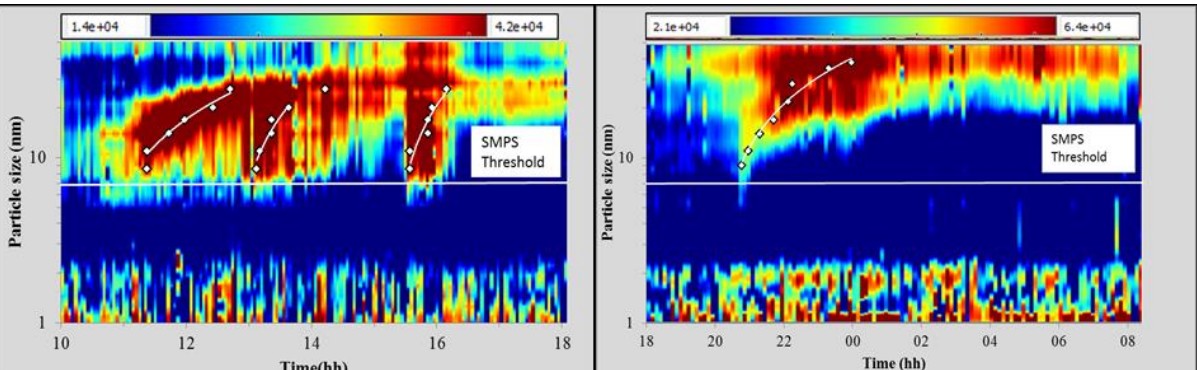


Figure 3: NAIS spectragrams of the growth events that occurred during (a) day time (b) night time.
Note the "floating banana" shape which indicates that these are clearly not NPF events. The SMPS
cannot detect particles at sizes below the horizontal white line. The colour contour represents the PNC
and the markers represent the times at which the PNC reached its maximum value at each particle size.
The unit of PNC is per cubic centimetre.





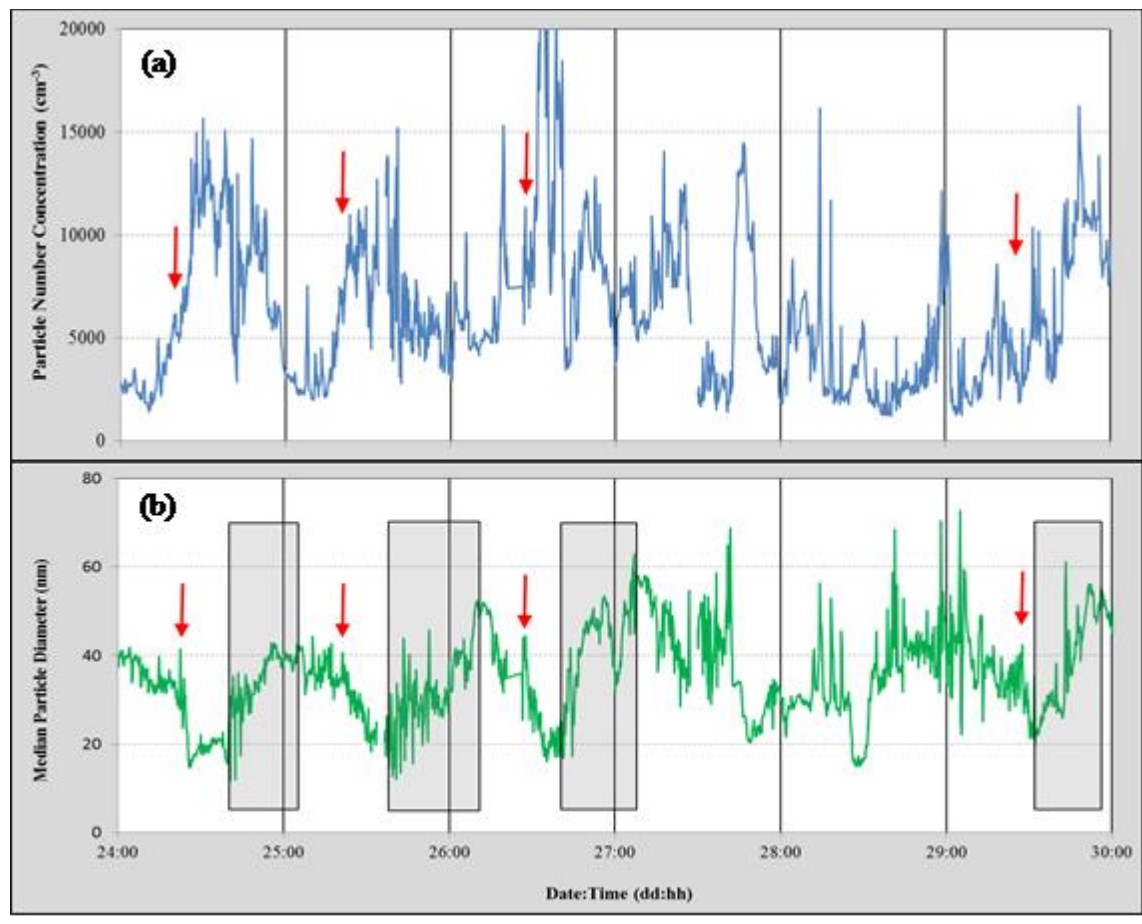


Figure 4: (a) the total PNC and (b) median particle diameter from the SMPS during 24 July-30 July,

2012. Red arrows and gray boxes represent the day time NPF events and the growth events,

respectively.









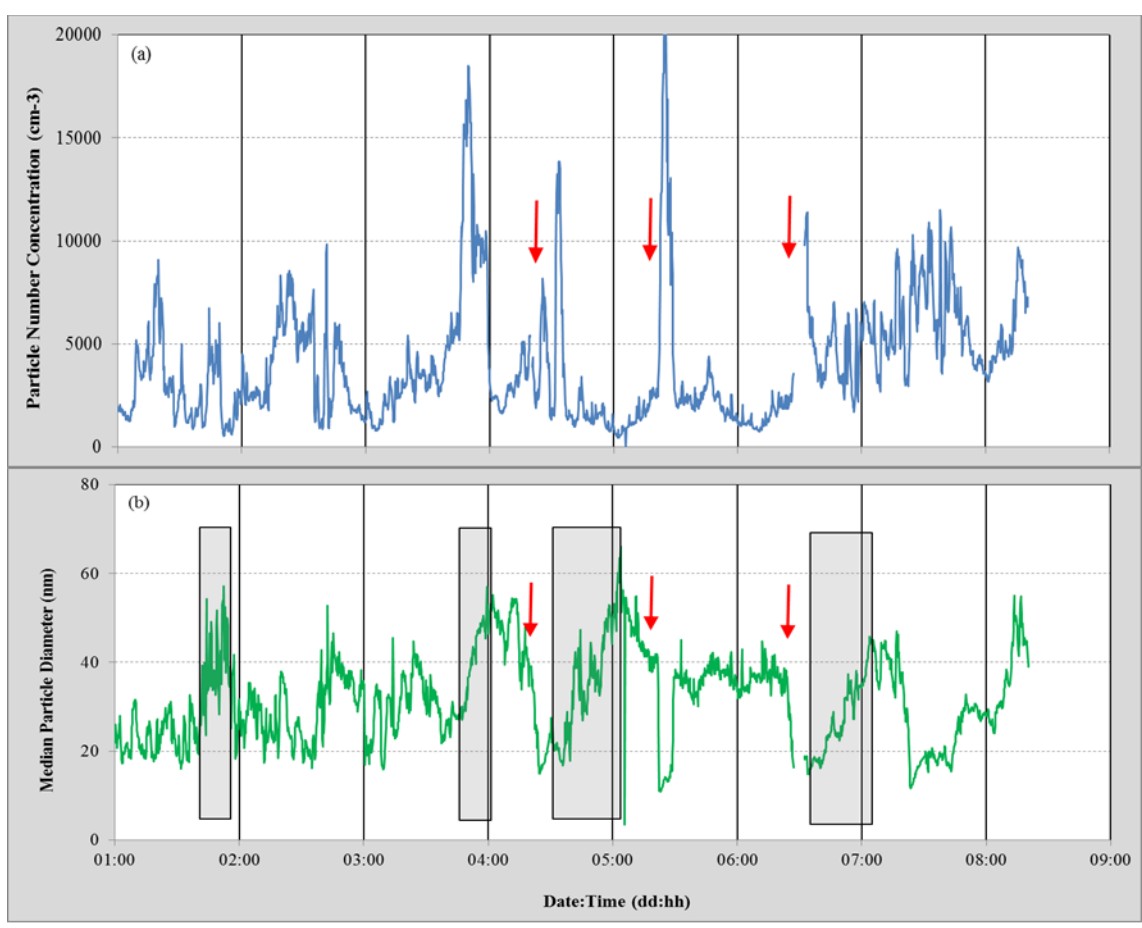

**Figure 5:** (a) the total PNC and (b) median particle diameter from the SMPS during 1 June-7 June, 2012. Red arrows and gray boxes represent the daytime NPF events and the growth events, respectively.

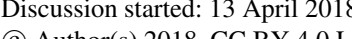



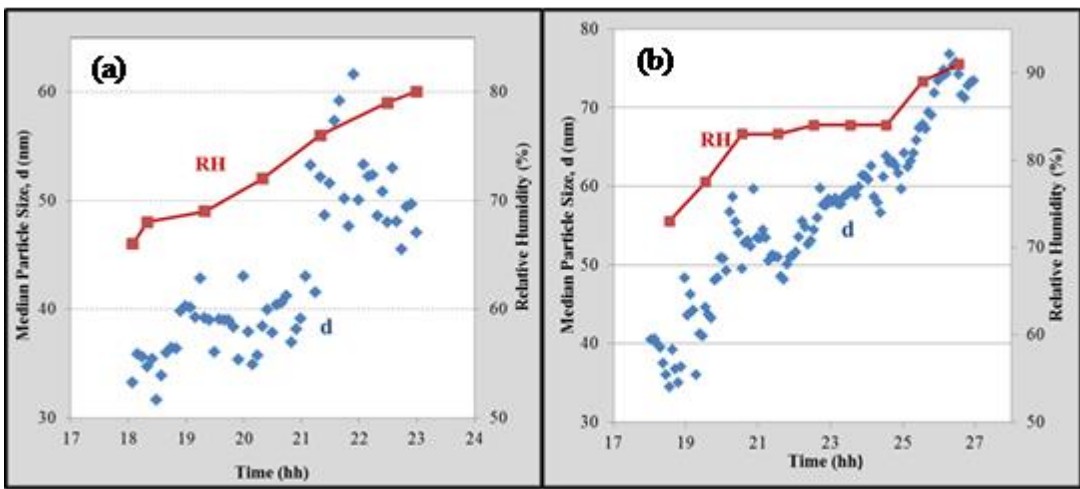


Figure 6: Median particle size and relative humidity as a function of time for growth events on July 16
and July 20, 2012, respectively






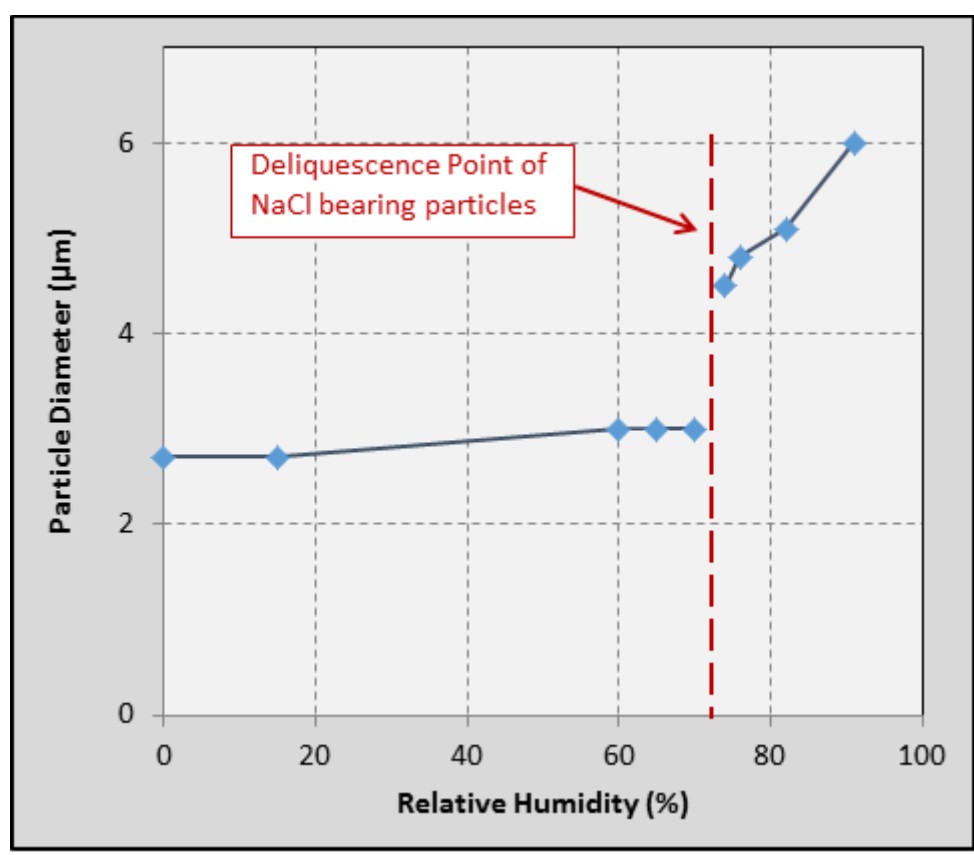



Figure 7: Particle diameter as a function of relative humidity for NaCl-bearing particles. The figure is
plotted using the data presented in Wise et al. (2007).









**Table 1: Summary of studies reporting night time NPF events**
SMPS: Scanning mobility particle sizer, AIS: Air ion spectrometer, BSMA: Balanced scanning mobility
analyser, FMPS: Fast mobility particle sizer

| Study | Location | Occurrence rate | | Instrument (size range) |
|---|---|---|---|---|
| | | **Day time** | **Night time** | |
| Svenningsson et al. (2008) | Abisko, Sweden (characterized by Subartic birch forest) | 46/195 days (23%) | 31/195 days (16%) | SMPS (10-500 nm) AIS (0.4-40 nm) |
| Junninen et al. (2008) | Pine Forest, Hyytiala, Finland | | 344/1279 days (27%) | BSMA (0.4-6.3nm) AIS (0.34-40 nm) |
| Suni et al. (2008) | Eucalypt forest, Tumbarumba, Australia | 184/351 days (52%) | 112/351days (32%) | AIS (0.34-40 nm) |
| Kalivitis et al. (2012) | Finokalia, Lassithiou, Greece (remote coastal site) | 53/365 days (15%) | 39/365 days (11%) | SMPS (9-900 nm) AIS (0.8-42 nm) |
| Man et al. (2015) | Suburban coastal site, Hong Kong | 12/112 days (11%) | 5/112 days (4%) | FMPS (5.6-560 nm) |
| Mazon et al. (2016) | SMEAR II, boreal forest, Hyytiala, Finland | | using neg ions: 1324/4015 days (34% ) | BSMA (0.8-8 nm) |
| | | | using pos ions: 1172 /4015 days (30%) | |



| Salimi et al. (2017) | 25 sites across Brisbane (characterized by urban environment) | 146/285 days (51%) | 73/285 days (26%) | SMPS (9-414 nm) |
|---|---|---|---|---|
| Kammer et al. (2017) | Landes forest, France | 2/16 days (12.5%) | 6/16 days (37.5%) | SMPS (10-478 nm) |























Table 2: Summary of the day time and night time NPF events

| Year | Total Data Available Days | Strong NPF events | | | Weak NPF events | | | Total NPF events | | |
|------|---------------------------|-------------------|------------|-------|-----------------|------------|-------|------------------|------------|-------|
|      |                           | Day time | Night time | Total | Day time | Night time | Total | Day time | Night time | Total |
| 2012 | 253 | 97 | 7 | 104 | 9 | 9 | 18 | 106 | 16 | 122 |
| 2015 | 65 | 18 | 4 | 22 | 5 | 7 | 12 | 23 | 11 | 34 |
| 2017 | 167 | 44 | 7 | 51 | 16 | 13 | 29 | 60 | 20 | 80 |
| Total Events |  | 159 | 18 | 177 | 30 | 29 | 59 | 189 | 47 | 236 |
| Total days | 485 | 159 | 18 | 177 | 30 | 29 | 59 | 181 | 47 | 213 |
| Occurrence rate (%) |  | 33 | 4 | 37 | 6 | 6 | 12 | 37 | 10 | 44 |

















Table 3: The mean and the range of meteorology and gas phase parameters on NPF and non-event days

| Parameter | Winter Months | | Summer Months | | NPF days | non-event days |
|---|---|---|---|---|---|---|
| | NPF days | non-event days | NPF days | non-event days | | |
| Meteorology | | | | | | |
| Solar radiation (Wm$^{-2}$) | 346 (230-490) | 316 (95-476) | 600 (202-818) | 476 (68-818) | 505 (202-818) | 397 (68-818) |
| Temperature ($^0$C) | 17 (12-19) | 16 (12-25) | 24 (18-29) | 24 (19-32) | 21 (12-29) | 20 (12-32) |
| Relative Humidity (%) | 59 (31-73) | 70 (27-90) | 52 (23-73) | 63 (25-86) | 54 (23-73) | 66 (25-90) |
| Wind direction ($^o$) | 215 S-SW | 203 S-SW | 197 S-SW | 177 S-SW | 205 S-SW | 200 S-SW |
| Wind Speed (ms$^{-2}$) | 0.38 (0.1-1.1) | 0.42 (0.1-1.3) | 0.57 (0.1-1.7) | 0.81 (0.1-2.1) | 0.5 (0.1-1.7) | 0.62 (0.1-2.1) |
| Gas Phase | | | | | | |
| Visibility (Mm$^{-1}$) | 15 (6-42) | 34 (2-112) | 19 (7-41) | 29 (6-114) | 18 (6-42) | 31 (2-114) |
| Ozone (ppb) | 12 (1-29) | 10 (2-26) | 20 (1-32) | 19 (3-35) | 17 (1-32) | 15 (2-35) |
| SO$_2$ (ppb) | 7 (6-10) | 6 (1-9) | 5 (1-14) | 3 (1-9) | 6 (1-14) | 5 (1-9) |






Table 4: Summary of the growth events, which did not follow the NPF events, obtained using the NAIS
data.

| Year | Total Data Available Days | Growth Events | | |
|------|------|------|------|------|
| | | Day time | Night time | Total |
| 2012 | 253 | 24 | 59 | 83 |
| 2015 | 65 | 4 | 21 | 25 |
| 2017 | 167 | 26 | 55 | 81 |
| | | | | |
| Total events | | 54 | 135 | 189 |
| Total days | 485 | 54 | 135 | 179 |
| Occurrence rate (%) | | 11 | 28 | 37 |




