# Peer review of "Differentiating between particle formation and growth events in an urban environment"

_Atmospheric Chemistry and Physics, 2018_

## Referee Comment (RC1) · Anonymous Referee #2 · 28 Jun 2018

This manuscript presents an analysis on new particle formation (NPF) and growth events based on extensive ambient measurements at an urban location. This is a valuable data set that should be published. However, in its current form the manuscript requires revisions, some of which can be considered substantial. My detailed comments in this regard are given below.

The last two paragraphs in section 3.4 give an impression that sub-10 nm particles might grow faster in this environment than larger particles. This is an interesting observation, if true. In most sites where ion spectrometers have been used for reported NPF studies, the particle growth rate was observed to increase from sub- 3 nm sizes up to 10-20 nm. I would like to see a bit more discussion on this topic in this paper, including comparison to earlier studies.

[Figure]

I am surprised how the authors ended up in selecting the few short-term campaigns when discussing particle growth following NPF in section 3.6 (lines 322-328). Growth to larger sizes occurs very frequently in so-called regional NPF events and in many locations, newly-formed particles have been observed to grow up to sizes where they may act as cloud condensation nuclei (50-150 nm in diameter). So growth following NPF is a very common phenomenon. The authors should bring this up more clearly in that paragraph, now the reader easily get a wrong impression that growth to larger sizes is kind of a rare phenomenon.

I am not comfortable with the last paragraph of section 3.6 (lines 343-358). By reading it, one easily gets an impression that water uptake alone might explain the observed particle growth at increasing RH. This is very unlikely to be the case. Firstly, comparison of the growing particles water uptake to that by NaCL is unfair, since the latter is perhaps the most hygroscopic material present in the ambient atmosphere, while ultra-fine particles in an urban environment are (based on measurements in several sites) much less hygroscopic. However, high RH might favor particle growth due to other reasons: 1) heterogenous reactions taking place in the liquid phase of the growing particles, or 2) simply due to the fact that an increase in RH is often accompanied by a decrease in ambient temperature, which would favor the transport of any semi-volatile compounds from the gas phase to these particles. I would recommend rewriting this paragraph and removing Figure 7 altogether.

In addition to the paragraphs mentioned above, there are many places in the text that lack references, either totally or proper/fresh ones: 1: line 42: the particle growth varies with particle size, 2) lines 57-58: Oxides of. . ., 3) lines 64-65: Numerous studies. . ., 4) the paragraph on lines 48-55: there are plenty of fresher papers on this, even reviews, that could be mentioned here.

Figure 1a seems unnessary to me, as all the required information can be obtained from figure 1b. I recommend removing figure 1a from the paper.

[Figure]

---

## Referee Comment (RC2) · Anonymous Referee #3 · 28 Jun 2018

The manuscript presents data from new particle formation (NPF) events in an urban environment in Brisbane, Australia. The main finding is that some NPF events could be misidentified growth of aerosol particles. The main issue is that this interpretation relies solely on the NAIS size distributions measured with the particle mode. The presented size distribution (Fig. 3) has no charger ion signal, which is an indication that the detection of the smallest size fraction was faulty. When the NAIS is adjusted to filter all the corona changer ions, it will also filter the newly formed particles measured within the particle mode. Nevertheless, I would not recommend to plot the corona ions a part of the particle spectra as it is not a real signal from ambient sample.

I would recommend that the authors check the diagnostic values for these days and contact the instrument vendor to determine the quality of the measurements. The

comparison of these results to the NAIS ion mode measurements is also extremely important, especially, to understand the performance of their instrument. Authors mention on Page 7, Line 153 that also the ion mode measurements were recorded.

The manuscript also does not reveal the source of the freshly formed 20 nm size particles that grow. It is visible in Figure 5 that the 2 growth only events correlate (at least the 2nd is clearly visible by eye) with an increase in particle number concentration. How well does your SMPS and NAIS agree in particle mode? Show a comparison figure as they overlapping size range.

Please find below a few line by line comments as the missing charger ions are clearly the largest issue at the moment (see comment to Fig. 3 below):

L360: Hard to see a growth until the early morning. Was the time series smoothed? The mean diameter seems to plateau.

L362-365: The description of the what is presented in Fig 5 is a bit limited. Also: what happened on the 5th June midday.

L367-374: Seems out of place in the result part, move this to the discussion or introduction?

L367-381: "Time" is the time of day?

L376-381: How does this look for NPF events?

L398: Factor of 2 not 1.

Comments to Figures:

Fig. 1: To see seasonality or the lack of it, the NPF frequency for each month should be shown.

Fig. 2: Data below 2 nm is from charger ions, so it's not a signal but an artifact.

Fig. 3: Why are there almost no charger ions (the signal below about ∼2nm) in this

[Figure]

figure? This could indicate a problem with the detection of small aerosol particles in the NAIS. Compare charger ions in Figure 3 to those in Figure 2.

Fig. 4 & 5: Was the time series smoothed? It looks rather noisy.

Fig. 6 & 7: The effect shown in Fig 7 is not visible in Fig 6. Plotting diameter vs RH in Figure 6 would allow immediate comparison with Fig 7.

[Figure]

---

## Author Comment (AC1) · 9 Jul 2018

Response to Anonymous Referee #1 Overall Comment The manuscript describes the difference between growth patterns during two main types of conditions:

(1) Right after a NPF event (2) without a NPF

This idea is actually interesting to consider.

Comment 1 In case this will be published in ACPD, I would love the authors to make a more extensive literature review about NPF studies in the urban atmospheres such as that reported in Japan, China, and different parts in the EU (particularly in Scandinavian countries and central Europe).

[Figure]

Response 1 While there are many papers reporting NPF in urban environments, there are a limited number of papers reporting night time NPF events. In Table 1, we have listed only those studies that have reported occurrence rates of NPF events BOTH during the day time and the night time.

Comment 2 Another important point that needs to be also resolved is the night time events and how the authors considered the start and end of night time. The authors took a fixed time for both sunrise and sunset. I would rather see this analysis to into account the real sun rise and sunset time.

Response 2 The method of estimating the start times of NPF events is as stated in Lines 141-146: "Every NPF event was characterised by a sharp increase of the PNC in the intermediate size range from 2.0-7.0 nm. This observation has been used to determine the starting time of an NPF event (Leino et al., 2016). Similarly, in the present study, the starting time of a strong NPF event was determined by noting the time of first occurrence of dN/dt > 10,000 cm-3h-1. The starting time of a weak NPF event was determined by noting the time of first occurrence of dN/dt > 5000 cm-3h-1. N is the number of particles in the size range 2.0-10 nm". In this paper, we did not consider the end times of NPF events. As suggested by the reviewer, we considered the real sunrise and sunset times and the results are shown in the revised Figure (1). Figure 1(a) shows the distribution of start times of daytime NPF events as a function of time after sunrise shown on the x-axis. The three bars on the extreme left correspond to times before sunrise. We have classified these as 'night time events'. Figure 1(b) shows the 'night time' events with time after sunset shown on the x-axis. We have deleted the original Figure 1(a) as suggested by Anonymous Referee #2 and replaced the original Figure 1(b) with these two new figures. Replacing real time with times after sunrise and sunset introduces quite a few changes to the description. To accommodate this, we will include the following new text into the paper: In Section 3.2, Lines 248-256: "Figures 1 (a) and (b) shows the summary of starting times of all NPF events during the day time and night time, respectively, estimated by using the method described

in Section 2.3.1. The histograms show the number of events observed in each 30 min period after sunrise and sunset, respectively. The times indicated on the x-axis refer to the end of each 30 min period. In Figure 1(a), the three bars at the extreme left correspond to times before sunrise. We have classified these as night time events. Both of these figures show that most NPF events (71%) began during the morning, with a high likelihood of occurrence between 2 and 4 hours after sunrise, corresponding to approximately between 8.00 am and 10.00 am". And in Lines 262-264: "The starting times of night time NPF events also showed a distinct trend with a peak likelihood between 3 and 4 hours after sunset, corresponding to approximately 8 and 9 pm". And, in Section Summary and Conclusions, Lines 474-477: "71% of NPF events occurred during the morning, with the highest probability of occurrence between 2 and 4 hours after sunrise, corresponding to approximately between 8.00 am and 10.00 am. Most of the night time events occurred between 3 and 4 hours after sunset, corresponding to approximately between 8.00 pm and 9:00 pm".

Response to Anonymous Referee #2 Overall Comments This manuscript presents an analysis on new particle formation (NPF) and growth events based on extensive ambient measurements at an urban location. This is a valuable data set that should be published. However, in its current form the manuscript requires revisions, some of which can be considered substantial. My detailed comments in this regard are given below.

Comment 1 The last two paragraphs in section 3.4 give an impression that sub-10 nm particles might grow faster in this environment than larger particles. This is an interesting observation, if true. In most sites where ion spectrometers have been used for reported NPF studies, the particle growth rate was observed to increase from sub-3 nm sizes up to 10-20 nm. I would like to see a bit more discussion on this topic in this paper, including comparison to earlier studies.

Response 1 We will insert the following text on section 3.4, Lines 333-349: "Typically, the particle growth rates were high during the first few hours and then decreased to a

few nanometres per hour within 3-4 hours after nucleation. Several studies have reported that the growth rate of particles in the size range 7-20 nm was greater than that in the smaller size range 3-7 nm (Backman et al., 2012;Gagné et al., 2011;Manninen et al., 2010;Yli-Juuti et al., 2009). Manninen et al. (2010) studied NPF events at 12 European sites and found that 9 out of the 12 sites showed this trend while at 3 sites the growth rate was greater in the smaller size range. They suggested that this size dependence was due to different condensing vapours participating in the growth of different sized particles depending on their saturation vapour pressures. For example, it is well known that sulfuric acid plays a dominant role in nucleation and the initial growth of particles during NPF while organics dominate the growth at larger sizes of 10-30 nm (Smith et al., 2008;Manninen et al., 2009;Yli-Juuti et al., 2011). Further evidence comes from the observation that the growth rate of the particles in the larger size range of 7-20 nm is enhanced during the summer when the concentration of biogenic volatile organic compounds in the atmosphere is greater (Yli-Juuti et al., 2011). Our observations of particle growth rates in the different size ranges agree with previous studies that have suggested that the dominant condensable vapour in Brisbane is probably sulfuric acid, with organics playing a secondary role (Crilley et al., 2014)".

Comment 2 I am surprised how the authors ended up in selecting the few short-term campaigns when discussing particle growth following NPF in section 3.6 (lines 322-328). Growth to larger sizes occurs very frequently in so-called regional NPF events and in many locations, newly-formed particles have been observed to grow up to sizes where they may act as cloud condensation nuclei (50-150 nm in diameter). So growth following NPF is a very common phenomenon. The authors should bring this up more clearly in that paragraph, now the reader easily get a wrong impression that growth to larger sizes is kind of a rare phenomenon.

Response 2 This is a misunderstanding. As stated in Section 2.1, "The measurements were carried out during the three calendar years 2012, 2015 and 2017, yielding 485 complete days of data". Figures 4 and 5 in Section 3.6 are just two examples of a few

days each. They are not short-term campaigns.

We will modify the first sentence of this paragraph as follows, Lines 391-392:

"Continued growth of particles following NPF events is a common phenomenon and has been reported by several other researchers".

Comment 3 I am not comfortable with the last paragraph of section 3.6 (lines 343-358). By reading it, one easily gets an impression that water uptake alone might explain the observed particle growth at increasing RH. This is very unlikely to be the case. Firstly, comparison of the growing particles water uptake to that by NaCL is unfair, since the latter is perhaps the most hygroscopic material present in the ambient atmosphere, while ultrafine particles in an urban environment are (based on measurements in several sites) much less hygroscopic. However, high RH might favor particle growth due to other reasons: 1) heterogenous reactions taking place in the liquid phase of the growing particles, or 2) simply due to the fact that an increase in RH is often accompanied by a decrease in ambient temperature, which would favor the transport of any semi-volatile compounds from the gas phase to these particles. I would recommend rewriting this paragraph and removing Figure 7 altogether.

Response 3 As suggested by the reviewer, we will remove Figure 7 along with its discussion, and modify the text in this paragraph to accommodate the comments. It will read as follows, Lines 413-435: "It is well-known that relative humidity may favour particle growth in the atmosphere owing to several reasons. For example, atmospheric aerosol particles increase in size with relative humidity due to the uptake of water (Winkler, 1988). In addition, when the relative humidity increases, heterogeneous reactions can take place in the liquid phase of a growing particle while, if there is an accompanying drop in temperature, it would enhance the transport of semivolatile compounds from the gas phase on to the surface of the particles. Water uptake is caused by the deliquescence of soluble salts which form a solution when the solid compound is exposed to water vapour at sufficiently high vapour pressure. Several organic materials

are also known to absorb water at high humidity which is more generally known as hygroscopicity. Sodium chloride (NaCl) has a deliquescence point of 76% relative humidity. At this point, a NaCl-bearing particle will deliquesce and become a solution of droplet with a well-defined spherical shape. The particle diameter does not change considerably as the relative humidity is increased from 0 to 74%, beyond which it can increase considerably. Close to the coast, sea-salt aerosols constitute a large proportion of the atmospheric particulate mass and NaCl is a major component. Many of the inorganic substances that readily absorb water, such as sea salt, ammonium salts and nitrates, are present in the Brisbane environment (Harrison, 2007). Therefore, it is not surprising that, in the present study, we observed that particle growth occurred on 7 out of 10 nights with high relative humidity".

Comment 4 In addition to the paragraphs mentioned above, there are many places in the text that lack references, either totally or proper/fresh ones: 1: line 42: the particle growth varies with particle size, 2) lines 57-58: Oxides of. . ., 3) lines 64-65: Numerous studies. . ., 4 the paragraph on lines 48-55: there are plenty of fresher papers on this, even reviews, that could be mentioned here.

Response 4 We will add the following references: Line 42: (Backman et al., 2012;Gagné et al., 2011;Manninen et al., 2010) Lines 57-58: (Harrison, 2007;Seinfeld and Pandis, 2006) Lines 64-65: (Seinfeld and Pandis, 2006;Suni et al., 2008;Man et al., 2015;Pushpawela et al., 2018) Lines 48-49: (Kulmala et al., 2004;Backman et al., 2012;Gagné et al., 2011;Manninen et al., 2010;Manninen et al., 2009;Rose et al., 2015) Line 53: (Birmili and Wiedensohler, 2000;Kulmala et al., 2004;Kulmala et al., 2013)

Comment 5 Figure 1a seems unnecessary to me, as all the required information can be obtained from figure 1b. I recommend removing figure 1a from the paper.

Response 5 As suggested by the reviewer, we will remove Figure 1(a) from the paper. In response to Reviewer #1, we will replace this figure with two figures showing the

day time and night time NPF events separately as a function of times after sunrise and sunset, respectively.

Response to Anonymous Referee #3 Comment 1 The manuscript presents data from new particle formation (NPF) events in an urban environment in Brisbane, Australia. The main finding is that some NPF events could be misidentified growth of aerosol particles. The main issue is that this interpretation relies solely on the NAIS size distributions measured with the particle mode. The presented size distribution (Fig. 3) has no charger ion signal, which is an indication that the detection of the smallest size fraction was faulty. When the NAIS is adjusted to filter all the corona changer ions, it will also filter the newly formed particles measured within the particle mode. Nevertheless, I would not recommend to plot the corona ions a part of the particle spectra as it is not a real signal from ambient sample. I would recommend that the authors check the diagnostic values for these days and contact the instrument vendor to determine the quality of the measurements. The comparison of these results to the NAIS ion mode measurements is also extremely important, especially, to understand the performance of their instrument. Authors mention on Page 7, Line 153 that also the ion mode measurements were recorded.

Response 1 We thank the reviewer for pointing this out. We are aware of the problem with the NAIS and the corona charger ions. We will insert the following text in the Methods section (Lines 144-149): "In the particle mode, it uses a corona needle to charge the particles. This leads to an inherent problem where the very small particles cannot be distinguished from the corona ions (Manninen et al., 2016). For this reason, we have restricted the lower detection limit in the particle mode to 2 nm". As suggested, we will remove the data below 2 nm from all the particle spectrograms (Figures 2 and Figure 3 have been revised accordingly). It is probable that the charger ion signal in Figure 3 is 'missing' because of the colour scale used in the diagram. When we adjust the colour scale, the cluster ion band shows up clearly as shown in the Figure 8(new).

Comment 2 The manuscript also does not reveal the source of the freshly formed

20 nm size particles that grow. It is visible in Figure 5 that the 2 growth only events correlate (at least the 2nd is clearly visible by eye) with an increase in particle number concentration. How well does your SMPS and NAIS agree in particle mode? Show a comparison figure as they overlapping size range. Please find below a few line by line comments as the missing charger ions are clearly the largest issue at the moment (see comment to Fig. 3 below):

Response 2 20 nm was the value of the count median diameter of the particles that were present in the atmosphere as measured by the SMPS. These are background particles that arise from a number of sources in the environment, mainly motor vehicle emissions. Manninen et al. (2016) have reported that the NAIS over-estimates the particle number concentration at sizes in the range 20-42 nm. Our results agreed with this observation with a difference of up to a factor of 20% in this size range. This did not create an issue as the particle number concentration in this size range was not quantified nor used in any formulations in this study. We accept that, this will affect the particle number concentrations in the larger sizes in the spectragrams shown in Figures 2 and 3. This was not a big issue as the particle number concentrations are represented in colour contours and were not quantified in these figures. The particle number concentrations in Figures 4, 5 and 6 are not from the NAIS – they are from the SMPS. So, there was no issue there.

Comment 3 L360: Hard to see a growth until the early morning. Was the time series smoothed? The mean diameter seems to plateau.

Response 3 This refers to Figure 4. There are 4 NPF events in this figure and they are indicated by the red arrows. The particle growth times are shaded in grey. The green line is the median particle diameter and this increases within all four grey shaded boxes. The right edge of each box indicates when the growth ends and these occur during the early hours of the day. If it would make it clearer, we will change "early morning" to "early hours".

The SMPS scans were obtained at time intervals of 5 min. The time series were not smoothed. The mean diameter (green line) does not plateau inside the grey shaded boxes.

Comment 4 L362-365: The description of the is presented in Fig 5 is a bit limited. Also: what happened on the 5th June midday.

Response 4 Figure 5 further supports what is presented in Figure 4. It shows two growth events that followed NPF events and an NPF event that was not followed by a growth event. 5th June midday appears to be a particle burst event. A sharp burst of new particles decreases the median particle size to less than 20 nm. The burst lasts for about 2 hours thereby eliminating the possibility of it being due to a motor vehicle emission plume or a person smoking a cigarette etc.

Comment 5 L367-374: Seems out of place in the result part, move this to the discussion or introduction?

Response 5 We do not have a separate discussion part and this paragraph is within the Results and Discussion section. This comparison follows our observations in the previous paragraph. To illustrate this more clearly, we will modify the first sentence as follows, Line 391-392: "These observations of continued growth of particles following NPF events is a common phenomenon and has been reported by several other researchers".

Comment 6 L367-381: "Time" is the time of day?

Response 6 Yes, it is the Time of Day. We will replace the x-axis titles "Time" in Figure 6 (a) and (b) with "Time of day".

Comment 7 L376-381: How does this look for NPF events?

Response 7 Relative humidity increases during the night time. Our observations show an increased growth rate with increasing relative humidity. We did not observe an effect of relative humidity on the frequency of NPF, perhaps because the relative humidity is

generally low during the day time. We have reported this in our earlier publications (Jayaratne et al., 2016;Pushpawela et al., 2018).

Comment 8 L398: Factor of 2 not 1.

Response 8 This sentence has now been removed in response to a comment by Reviewer #2.

Comments to Figures: Fig. 1: To see seasonality or the lack of it, the NPF frequency for each month should be shown.

Response The scarcity of data during some months, notably January to March, prevented us from deriving a reliable seasonal distribution chart. Figure 9 (new) is the chart including all months with at least 10 observational days: In figure 9 (new), note that the data spans three calendar years. However, the data for April was from just one year. Hence, the unusually high percentage. The data for December is also from just one year. During this time there were a number of controlled burning events around Brisbane and this accounted for the low percentage of NPF events. The seasonal dependence chart, in our opinion, is not extensive enough to be presented in this paper.

Fig. 2: Data below 2 nm is from charger ions, so it's not a signal but an artifact.

Response The graphs have been revised and now show only the values above 2 nm (See response to Comment 1 above).

Fig. 3: Why are there almost no charger ions (the signal below about 2nm) in this figure? This could indicate a problem with the detection of small aerosol particles in the NAIS. Compare charger ions in Figure 3 to those in Figure 2.

Response See response to Comment 1 above.

Fig. 4 & 5: Was the time series smoothed? It looks rather noisy.

Response As stated in Response 3, the SMPS scans were obtained at time intervals of 5 min. The time series were not smoothed. The noise is normal in an urban environment where the dominant source of aerosols, particularly ultrafine particles, is from motor vehicles.

Fig. 6 & 7: The effect shown in Figure 7 is not visible in Figure 6. Plotting diameter vs RH in Figure 6 would allow immediate comparison with Figure 7.

Response In response to Reviewer#2 (Comment 3), we will remove Figure 7 from the paper. As such, there is no need for a comparison with Figure 6.

[revised manuscript text omitted]

---

## Author Response (AR1)

**Professor Lidia Morawska**
International Laboratory for Air Quality and Health
Queensland University of Technology
George Street, Brisbane QLD 4001 Australia
Email: l.morawska@qut.edu.au

5th July 2018

Natascha Töpfer
Copernicus Publications
Editorial Support
editorial@copernicus.org

Dear Natascha,

**Submission of Revised Manuscript Number: acp-2018-189**

**Title: Differentiating between particle formation and growth events in an urban environment**

**Authors (names and email addresses):**

Ms. Buddhi Pushpawela: buddhi.pushpawela@hdr.qut.edu.au
Dr. Rohan Jayaratne: r.jayaratne@qut.edu.au
Prof. Lidia Morawska: l.morawska@qut.edu.au

As requested, we have considered the comments of the three anonymous reviewers in detail and revised the paper accordingly.

I am submitting the following documents:

(1) Revised Manuscript (2) Revised Manuscript with all changes indicated in Track Changes (3) Detailed responses to Anonymous Reviewers 1, 2 and 3.

I hope you will find it acceptable for publication in ACP.

Please contact me at the email address below, should you have any further queries.

Yours sincerely,

**Professor Lidia Morawska, PhD**

**Director**
International Laboratory for Air Quality and Health
WHO CC for Air Quality and Health

**Director - Australia**
Australia – China Centre for Air Quality Science and Management
Queensland University of Technology
Phone: +61 7 3138 2616
Fax: +61 7 3138 9079
E-mail: l.morawska@qut.edu.au

**Response to Anonymous Referee #1**

Overall Comment

The manuscript describes the difference between growth pattern during two main types of conditions:

(1)      Right after a NPF event (2)      without a NPF

This idea is actually interesting to consider.

Comment 1

In case this will be published in ACPD, I would love the authors to make a more extensive literature review abut NPF studies in the urban atmospheres such as that reported in Japan, China, and different parts in the EU (particularly in Scandinavian countries and central Europe).

Response 1

While there are many papers reporting NPF in urban environments, there are a limited number of papers reporting night time NPF events. In Table 1, we have listed only those studies that have reported occurrence rates of NPF events BOTH during the day time and the night time.

Comment 2

Another important point that needs to be also resolved is the night tie events and how the authors considered the start and end of night time. The authors took a fixed time for both sunrise and sunset. I would rather see this analysis to into account the real sun rise and sunset time.

Response 2

The method of estimating the start times of NPF events is as stated in Lines 141-146:

*"Every NPF event was characterised by a sharp increase of the PNC in the intermediate size range from 2.0-7.0 nm. This observation has been used to determine the starting time of an NPF event (Leino et al., 2016). Similarly, in the present study, the starting time of a strong NPF event was determined by noting the time of first occurrence of $dN/dt > 10,000$ $cm^{-3}h^{-1}$. The starting time of a weak NPF event was determined by noting the time of first occurrence of $dN/dt > 5000$ $cm^{-3}h^{-1}$. N is the number of particles in the size range 2.0-10 nm".*

In this paper, we did not consider the end times of NPF events.

We considered the real sunrise and sunset times and the results are shown in the two figures below. Figure 1(a) shows the 'daytime' events with time after sunrise shown on the x-axis.

[Figure]

Figure 1(a): Distribution of start times of daytime NPF events as a function of time after sunrise.

The three bars on the extreme left correspond to times before sunrise. We have classified these as 'night time events'.

Figure 1(b) shows the 'night time' events with time after sunset shown on the x-axis.

[Figure]

Figure 1(b): Distribution of start times of night time NPF events as a function of time after sunset.

We have deleted the original Figure 1(a) as suggested by Anonymous Referee #2 and replaced the original Figure 1(b) with these two new figures.

Replacing real time with times after sunrise and sunset introduces quite a few changes to the description. To accommodate this, we have included the following new text into the paper:

In Section 3.2, Lines 248-256:

*"Figures 1 (a) and (b) shows the summary of starting times of all NPF events during the day time and night time, respectively, estimated by using the method described in Section 2.3.1. The histograms show the number of events observed in each 30 min period after sunrise and sunset, respectively. The times indicated on the x-axis refer to the end of each 30 min period. In Figure 1(a), the three bars at the extreme left correspond to times before sunrise. We have classified these as night time events. Both of these figures show that most NPF events (71%) began during the morning, with a high likelihood of occurrence between 2 and 4 hours after sunrise, corresponding to approximately between 8.00 am and 10.00 am".*

And in Lines 262-264:

*"The starting times of night time NPF events also showed a distinct trend with a peak likelihood between 3 and 4 hours after sunset, corresponding to approximately 8 and 9 pm".*

And, in Section 474-477, Line(Summary and Conclusions):

*"71% of NPF events occurred during the morning, with the highest probability of occurrence between 2 and 4 hours after sunrise, corresponding to approximately between 8.00 am and 10.00 am. Most of the night time events occurred between 3 and 4 hours after sunset, corresponding to approximately between 8.00 pm and 9:00 pm".*

**Response to Anonymous Referee #2**

Overall Comments

This manuscript presents an analysis on new particle formation (NPF) and growth events based on extensive ambient measurements at an urban location. This is a valuable data set that should be published. However, in its current form the manuscript requires revisions, some of which can be considered substantial. My detailed comments in this regard are given below.

Comment 1

The last two paragraphs in section 3.4 give an impression that sub-10 nm particles might grow faster in this environment than larger particles. This is an interesting observation, if true. In most sites where ion spectrometers have been used for reported NPF studies, the particle growth rate was observed to increase from sub- 3 nm sizes up to 10-20 nm. I would like to see a bit more discussion on this topic in this paper, including comparison to earlier studies.

Response 1

We have inserted the following text on section 3.4 Lines 333-349:

*"Typically, the particle growth rates were high during the first few hours and then decreased to a few nanometres per hour within 3-4 hours after nucleation. Several studies have reported that the growth rate of particles in the size range 7-20 nm was greater than that in the smaller size range 3-7 nm (Backman et al., 2012; Gagne et al., 2011; Manninen et al., 2010; Yli Juuti et al., 2009). Manninen et al (2010) studied NPF events at 12 European sites and found that 9 out of the 12 sites showed this trend while at 3 sites the growth rate was greater in the smaller size range. They suggested that this size dependence was due to different condensing vapours participating in the growth of different sized particles depending on their saturation vapour pressures. For example, it is well known that sulfuric acid plays a dominant role in nucleation and the initial growth of particles during NPF while organics dominate the growth at larger sizes of 10-30 nm (Smith et al, 2008; Manninen et al., 2009; Yli-Juuti et al., 2011). Further evidence comes from the observation that the growth rate of the particles in the larger size range of 7-20 nm is enhanced during the summer when the concentration of biogenic volatile organic compounds in the atmosphere is greater (Yli-Juuti et al., 2011). Our observations of particle growth rates in the different size ranges agree with previous studies that have suggested that the dominant condensable vapour in Brisbane is probably sulfuric acid, with organics playing a secondary role (Crilley et al, 2014)".*

Comment 2

I am surprised how the authors ended up in selecting the few short-term campaigns when discussing particle growth following NPF in section 3.6 (lines 322-328). Growth to larger sizes occurs very frequently in so-called regional NPF events and in many locations, newly-formed particles have been observed to grow up to sizes where they may act as cloud condensation nuclei (50-150 nm in diameter). So growth following

NPF is a very common phenomenon. The authors should bring this up more clearly in that paragraph, now the reader easily get a wrong impression that growth to larger sizes is kind of a rare phenomenon.

Response 2

As stated in Section 2.1, *"The measurements were carried out during the three calendar years 2012, 2015 and 2017, yielding 485 complete days of data".* Figures 4 and 5 in Section 3.6 are just two examples of a few days each. They are not short-term campaigns.

We have modified the first sentence of this paragraph as follows , Lines 391-392:

*"Continued growth of particles following NPF events is a common phenomenon and has been reported by several other researchers".*

Comment 3

I am not comfortable with the last paragraph of section 3.6 (lines 343-358). By reading it, one easily gets an impression that water uptake alone might explain the observed particle growth at increasing RH. This is very unlikely to be the case. Firstly, comparison of the growing particles water uptake to that by NaCL is unfair, since the latter is perhaps the most hygroscopic material present in the ambient atmosphere, while ultrafine particles in an urban environment are (based on measurements in several sites)

much less hygroscopic. However, high RH might favor particle growth due to other reasons: 1) heterogenous reactions taking place in the liquid phase of the growing particles, or 2) simply due to the fact that an increase in RH is often accompanied by a decrease in ambient temperature, which would favor the transport of any semi-volatile compounds from the gas phase to these particles. I would recommend rewriting this paragraph and removing Figure 7 altogether.

Response 3

We have removed Figure 7 along with its discussion, as suggested, and modified the text in this paragraph to accommodate the comments of the reviewer. It now reads as follows, Lines 413-435:

*"It is well-known that relative humidity may favour particle growth in the atmosphere owing to several reasons. For example, atmospheric aerosol particles increase in size with relative humidity due to the uptake of water (Winkler, 1988). In addition, when the relative humidity increases, heterogeneous reactions can take place in the liquid phase of a growing particle while, if there is an accompanying drop in temperature, it would enhance the transport of semivolatile compounds from the gas phase on to the surface of the particles. Water uptake is caused by the deliquescence of soluble salts which form a solution when the solid compound is exposed to water vapour at sufficiently high vapour pressure. Several organic materials are also known to absorb water at high humidity which is more generally known as hygroscopicity. Sodium chloride (NaCl) has a deliquescence point of 76% relative humidity. At this point, a NaCl-bearing particle will deliquesce and become a solution of droplet with a well-defined spherical shape. The particle diameter does not change considerably as the relative humidity is increased from 0 to 74%, beyond which it can increase considerably. Close to the coast, sea-salt aerosols constitute a large proportion of the atmospheric particulate mass and NaCl is a major component. Many of the inorganic substances that readily absorb water, such as sea salt, ammonium salts and nitrates, are present in the Brisbane environment (Harrison, 2007). Therefore, it is not surprising that, in the present study, we observed that particle growth occurred on 7 out of 10 nights with high relative humidity".*

Comment 4

In addition to the paragraphs mentioned above, there are many places in the text that lack references, either totally or proper/fresh ones: 1: line 42: the particle growth varies with particle size, 2) lines 57-58: Oxides of. . ., 3) lines 64-65: Numerous studies. . ., 4)

the paragraph on lines 48-55: there are plenty of fresher papers on this, even reviews, that could be mentioned here.

Response 4

The following references have been added:

Line 42: *Backman et al., 2012; Gagne et al., 2011; Manninen et al., 2010*

Lines 57-58: *Harrison, 2007; Seinfeld and Pandis, 2006*

Lines 64-65: *; Seinfeld and Pandis, 2006; Suni et al., 2008; Man et al., 2015; Pushpawela et al., 2018*

Lines 48-49: *Kulmala et al., 2004; Backman et al., 2012; Gagne et al., 2011; Manninen et al., 2009, 2010; Rose et al., 2015; Siingh et al., 2013*

Line 53: *Birmili and Wiedensohler, 2000; Kulmala et al., 2004, 2013*

Comment 5

Figure 1a seems unnessary to me, as all the required information can be obtained from figure 1b. I recommend removing figure 1a from the paper.

Response 5

We have removed Figure 1(a) from the paper.

In response to Reviewer #1, we have replaced this figure with two figures showing the day time and night time NPF events separately as a function of times after sunrise and sunset, respectively.

**Response to Anonymous Referee #3**

Comment 1

The manuscript presents data from new particle formation (NPF) events in an urban environment in Brisbane, Australia. The main finding is that some NPF events could be misidentified growth of aerosol particles. The main issue is that this interpretation relies solely on the NAIS size distributions measured with the particle mode. The presented size distribution (Fig. 3) has no charger ion signal, which is an indication that the detection of the smallest size fraction was faulty. When the NAIS is adjusted to filter all the corona changer ions, it will also filter the newly formed particles measured within the particle mode. Nevertheless, I would not recommend to plot the corona ions a part of the particle spectra as it is not a real signal from ambient sample.

I would recommend that the authors check the diagnostic values for these days and contact the instrument vendor to determine the quality of the measurements. The comparison of these results to the NAIS ion mode measurements is also extremely important, especially, to understand the performance of their instrument. Authors mention on Page 7, Line 153 that also the ion mode measurements were recorded.

Response 1

We thank the reviewer for pointing this out.

We are aware of the problem with the NAIS and the corona charger ions. We have inserted the following text in the Methods section (Lines 144-149):

*"In the particle mode, it uses a corona needle to charge the particles. This leads to an inherent problem where the very small particles cannot be distinguished from the corona ions (Manninen et al., 2016). For this reason, we have restricted the lower detection limit in the particle mode to 2 nm".*

As suggested, we have also removed the data below 2 nm from all the particle spectragrams (Figures 2 and 3 have been revised accordingly).

It is probable that the charger ion signal in Figure 3 is 'missing' because of the colour scale used in the diagram. When we adjust the colour scale, the cluster ion band shows up clearly as shown in the figure below:

[Figure]

As suggested, we have removed the data below 2 nm and redrawn this figure and it now appears as follows:

[Figure]

Comment 2

The manuscript also does not reveal the source of the freshly formed 20 nm size particles that grow. It is visible in Figure 5 that the 2 growth only events correlate (at least the 2nd is clearly visible by eye) with an increase in particle number concentration.

How well does your SMPS and NAIS agree in particle mode? Show a comparison figure as they overlapping size range.

Please find below a few line by line comments as the missing charger ions are clearly the largest issue at the moment (see comment to Fig. 3 below):

Response 2

nm was the value of the count median diameter of the particles that were present in the atmosphere as measured by the SMPS. These are background particles that arise from a number of sources in the environment, mainly motor vehicle emissions.

Manninen et al (2016) have reported that the NAIS over-estimates the particle number concentration at sizes in the range 20-42 nm. Our results agreed with this observation with a difference of up to a factor of 20% in this size range. This did not create an issue as the particle number concentration in this size range was not quantified nor used in any formulations in this study. We accept that, this will affect the particle number concentrations in the larger sizes in the spectragrams shown in Figures 2 and 3. This was not a big issue as the particle number concentrations are represented in colour contours and were not quantified in these figures. The particle number concentrations in Figures 4, 5 and 6 are not from the NAIS – they are from the SMPS. So, there was no issue there.

Comment 3

L360: Hard to see a growth until the early morning. Was the time series smoothed?

The mean diameter seems to plateau.

Response 3

This refers to Figure 4. There are 4 NPF events in this figure and they are indicated by the red arrows. The particle growth times are shaded in grey. The green line is the median particle diameter and this increases within all four grey shaded boxes. The right edge of each box indicates when the growth ends and these occur during the early hours of the day. If it would make it clearer, we have changed "*early morning*" to "*early hours*".

The SMPS scans were obtained at time intervals of 5 min. The time series were not smoothed.

The mean diameter (green line) does not plateau inside the grey shaded boxes.

Comment 4

L362-365: The description of the what is presented in Fig 5 is a bit limited. Also: what happened on the 5th June midday.

Response 4

Figure 5 further supports what is presented in Figure 4. It shows two growth events that followed NPF events and an NPF event that was not followed by a growth event.

5[th] June midday appears to be a particle burst event. A sharp burst of new particles decreases the median particle size to less than 20 nm. The burst lasts for about 2 hours thereby eliminating the possibility of it being due to a motor vehicle emission plume or a person smoking a cigarette etc.

Comment 5

L367-374: Seems out of place in the result part, move this to the discussion or introduction?

Response 5

We do not have a separate discussion part and this paragraph is within the Results and Discussion section. This comparison follows our observations in the previous paragraph. To illustrate this more clearly, we have modified the first sentence as follows, Line 391-392:

*"These observations of continued growth of particles following NPF events is a common phenomenon and has been reported by several other researchers".*

Comment 6

L367-381: "Time" is the time of day?

Response 6

Yes, it is the Time of Day.

We have replaced the x-axis titles "Time" in Figure 6 (a) and (b) with "Time of day".

Comment 7

L376-381: How does this look for NPF events?

Response 7

Relative humidity increases during the night time. Our observations show an increased growth rate with increasing relative humidity. We did not observe an effect of relative humidity on the frequency of NPF, perhaps because the relative humidity is generally low during the day time. We have reported this in our earlier publications (Jayaratne et al, 2016; Pushpawela et al., 2018).

Comment 8

L398: Factor of 2 not 1.

Response 8

This sentence has now been removed in response to a comment by Reviewer #2.

Comments to Figures:

Fig. 1: To see seasonality or the lack of it, the NPF frequency for each month should be shown.

Response

The scarcity of data during some months, notably January to March, prevented us from deriving a reliable seasonal distribution chart. Shown below is the chart including all months with at least 10 observational days:

[Figure]

Note that the data spans three calendar years. However, the data for April was from just one year. Hence, the unusually high percentage. The data for December is also from just one year. During this time there were a number of controlled burning events around Brisbane and this accounted for the low percentage of NPF events. The seasonal dependence chart, in our opinion, is not reliable enough to be presented in this paper.

Fig. 2: Data below 2 nm is from charger ions, so it's not a signal but an artifact.

Response

The graphs have been revised and now show only the values above 2 nm (See response to Comment 1 above).

Fig. 3: Why are there almost no charger ions (the signal below about 2nm) in this figure? This could indicate a problem with the detection of small aerosol particles in the NAIS. Compare charger ions in Figure 3 to those in Figure 2.

Response

See response to Comment 1 above.

Fig. 4 & 5: Was the time series smoothed? It looks rather noisy.

Response

As stated in Response 3, the SMPS scans were obtained at time intervals of 5 min. The time series were not smoothed. The noise is normal in an urban environment where the dominant source of aerosols, particularly ultrafine particles, are from motor vehicles.

Fig. 6 & 7: The effect shown in Fig 7 is not visible in Fig 6. Plotting diameter vs RH in Figure 6 would allow immediate comparison with Fig 7.

Response

In response to Reviewer#2 (Comment 3), we have removed Figure 7 from the paper. As such, there is no need for a comparison with Fig 6.

[revised manuscript text omitted]